# Interface Calculation of In Situ Micro-Nano TaC/NbC Ceramic Particle Composites

**DOI:** 10.3390/ma16051887

**Published:** 2023-02-24

**Authors:** Jilin Li, Yunhua Xu, Wanying Li

**Affiliations:** 1School of Materials Science and Engineering, Xi’an University of Technology, Xi’an 710048, China; 2Institute of New Materials, Guangdong Academy of Sciences, Guangzhou 510650, China; 3Guangzhou Yiyin New Materials Technology Co., Ltd., Guangzhou 510650, China

**Keywords:** interface binding energy, reaction in situ, interface computing, micro nano ceramics

## Abstract

Traditional experiments are difficult to accurately and quantitatively measure the interfacial properties of composites, such as interfacial bonding strength, interfacial microelectronic structure, and other information. It is particularly necessary to carry out theoretical research for guiding the interface regulation of Fe/MCs composites. In this research, the first-principles calculation method is used to systematically study the interface bonding work; however, in order to simplify the first-principle calculation of the model, dislocation is not considered in this paper, including interface bonding characteristics and electronic properties of α-Fe- and NaCl-type transition metal carbides (Niobium Carbide (NbC) and Tantalum Carbide (TaC)). The interface energy is related to the bond energy between the interface Fe atoms, C atoms and metal M atoms, and the interface energy Fe/TaC < Fe/NbC. The bonding strength of the composite interface system is accurately measured, and the interface strengthening mechanism is analyzed from the perspectives of atomic bonding and electronic structure, which provides a scientific guiding ideology for regulating the interface structure of composite materials.

## 1. Introduction

It is well-known that the precipitation of second-phase particles plays a crucial role in improving the strength of materials. In the second-phase strengthening, the external obstacles improve the strength of the material by dislocation pinning and hindering dislocation movement. In recent years, the second-phase strengthening of iron matrix has attracted more and more attention. Due to its high hardness, elastic modulus, melting point and thermal conductivity [1,2,3], and good compatibility with iron matrix, transition metal carbides (MCs) such as NbC and TaC are considered to be more effective second-phase reinforcements in iron matrix.

Park et al. [4] studied the interface between BCC-Fe and (V, Nb, Mo, W, and Ti)C carbides. The results show that the formation of composite carbides at the Fe/TiC interface can greatly reduce the interface energy of the composite, thereby improving the stability of the material. Jung et al. [5] calculated the interface energy between FCC-Fe and TiC, ZrC, VC, NbC, and TaC. The interface energy of the coherent interface is lower than that of the semi-coherent interface and VC is the most easily precipitated carbide among the five carbides. Chung et al. [1] calculated the interface energy of three transition metal carbides-reinforced BCC-Fe composites, and further confirmed that the interface energy is related to the bond energy between Fe atoms, C atoms, and metal M atoms, and the interface energy decreases in the order of Fe/VC < Fe/TaC < Fe/NbC.

Controlling the size and density of the second phase in the iron matrix is of great significance to improve the mechanical properties of iron, whether the second phase can become a pinned particle or a transition nucleation position also depends on the interface energy between the iron matrix and the second phase [5]. Therefore, an in-depth study of the interfacial properties between iron matrix and transition metal carbides has a crucial impact on improving the comprehensive mechanical properties of composites.

However, it is difficult to accurately measure the interfacial properties of composite materials by traditional experimental methods, such as interfacial bonding strength, interfacial microelectronic structure, and other information. Therefore, it is urgent to carry out theoretical research for guiding the interface regulation of Fe/MCs composites. Through investigation, it is found that first-principles calculations based on density functional theory are widely used to study the interface between Fe and carbides [6,7,8], which can accurately measure the bonding strength of the composite interface system and analyze the interface strengthening mechanism from the perspectives of atomic bonding and electronic structure to provide scientific guiding ideology for regulating the interface structure of composite materials. According to the literature, the matching of the interface of Fe- and NaCl-type transition carbides conforms to the Baker–Nutting relationship [9,10]: {100}_MCS_//{100}_Fe_, <110>_MCS_//<110>_Fe_. Based on this, the interfacial bonding work, interfacial bonding characteristics, and electronic properties of α-Fe- and NaCl-type transition metal carbides (NbC and TaC) were systematically studied by first-principles calculations.

## 2. Methods and Details

The CASTEP software package based on density functional theory is used. Since the GGA Generalized Gradient Approximation (GGA) can accurately predict the lattice constants, binding energies, and magnetic properties of 3d transition metals and their alloys, the PBE functional (Perdew–Burke–Enzerhof) in the generalized gradient approximation is used for the electron exchange correlation energy of each system in this study [11]. At the same time, Ultra Soft Pseudopotentials (USPPs) are used to describe the interaction force between valence electrons and ionic cores [12]. The whole structure is relaxed by the Broyden–Fletcher–Goldfarb–Shannon (BFGS) algorithm, the unit cell and atomic coordinate positions are optimized to reach the ground state [13]. The Fe, Nb, Ta, and C atoms are pseudo-pseudopotential-treated, and the extranuclear electrons are Fe 3d^6^4s^2^, Nb 4d^4^5s^1^, Ta 5d^3^6s^2^, and C 2s^2^2p^2^. The surface and interface models are constructed with periodic boundary conditions [14]. The Brillouin zone is integrated by the Monkhorst–Pack method [15]. The k-point values of the bulk phase, surface, and interface are 10 × 10 × 10, 6 × 6 × 2, 6 × 6 × 1. The plane wave cutoff energy is set to 360eV, and the self-consistent convergence accuracy is 1 × 10^−6^ eV/atom. The calculation results are analyzed and discussed by this method.

The interfaces studied in this paper were constructed by the following steps:An eleven-layer MCs (TaC/NbC) (1 0 0) surface was built, the termination can be either C or Ta/Nb, and in order to avoid creating the dipoles in the periodic cell artificially, both the lower and upper surfaces of the slab must be terminated with the same atomic specie, namely C or Ta/Nb atom;Similarly, we also cut an Fe (1 0 0) surface from the bulk Fe crystal with body centered cubic structure (BCC), the thickness of Fe slab was selected so that the interior of the slab is bulk-like, we found the slab with 5 atomic layers was sufficient; The model of Fe/ MCs interface uses a superlattice geometry in which a eleven-layer MCs (1 0 0) slab is placed between two five-layer slabs of Fe (1 0 0), resulting in two identical interfaces per supercell. Due to the periodic boundary condition, the interface structure contains eleven-layer slabs of Fe (1 0 0);In order to match the lattice geometry, the (1 0 0) surface of MCs was rotated to a set of new orientations, and then we placed the MCs slab in the middle between two Fe slabs. The orientation relationship we studied in this paper is Fe [0 0 1](1 1 0)//MCs [0 0 1](1 1 0).

## 3. Calculation Method (Results) and Discussion

### 3.1. Bulk Phase Characteristics of α-Fe and Carbide MCs

The crystal structures of α-Fe, NbC, and TaC are shown in Figure 1, where the crystal structures of α-Fe, NbC, and TaC are all cubic, with space groups corresponding to $\mathrm{IM}\bar{3}M$, $\mathrm{FM}\bar{3}M$, and $\mathrm{FM}\bar{3}M$. The bulk stability can be evaluated by the enthalpy of formation Δ*r*H. The enthalpy of formation is negative, and the smaller the value, the higher the stability of the bulk phase. Taking NbC as an example, the calculation formula of enthalpy of formation Δ*r*H is shown as follows:(1)$$\Delta_{r}H\left(\mathrm{NbC}\right)=E_{coh}\left(\mathrm{NbC}\right)-E_{coh}\left(\mathrm{Nb}\right)-E_{coh}\left(C\right),$$
(2)$$E_{coh}\left(\mathrm{NbC}\right)=E_{tot}\left(\mathrm{NbC}\right)-E_{iso}\left(\mathrm{Nb}\right)-E_{iso}\left(C\right),$$
where *E_coh_*(NbC), *E_coh_*(Nb) and *E_coh_*(C) are the binding energies of bulk NbC, Nb, and C atoms. *E_tot_* (NbC) is the total energy of NbC; *E_iso_*(Nb) and *E_iso_*(C) are the total energy of Nb and C atoms. Table 1 lists the lattice constants, volume, bulk modulus, enthalpy of formation, and their references for Fe, NbC, and TaC after geometric optimization.

It can be seen from the table for NbC and TaC, the enthalpy of formation is negative, indicating that the crystal structure has high thermodynamic stability. Secondly, the absolute value of TaC formation enthalpy is higher than that of NbC, and the thermodynamic stability of TaC bulk phase is higher. At the same time, taking NbC as an example, the three elastic constants (*C*_11_ = 578.53 GPa, *C*_12_ = 160.17 GPa, *C*_44_ = 155.77 GPa) of NbC all satisfy the generalized standard of stable crystals (*C*_11_ > 0, *C*_44_ > 0, *C*_11_–*C*_12_ > 0, *C*_11_ + 2*C*_12_ > 0), which are thermodynamically stable configurations. In addition, the lattice constant, bulk modulus, and enthalpy of formation calculated in this study are in good agreement with the reference values [16,17,18,19,20,21,22,23,24] and the error is within 5%. Therefore, the calculation methods and parameters used in this paper have high enough credibility to ensure the accuracy of subsequent composite surface and interface calculations.

**Table 1 materials-16-01887-t001:** Lattice constants (*a*, *b*, *c*), crystal volume (*V*), bulk modulus (*B*), enthalpy of formation (Δ*r*H) of α-Fe, NbC, and TaC and their reference values.

Compounds	*a*/*b*/*c* (Å)	*V* (Å^3^)	*B* (GPa)	Δ_r_H (eV·atom^−1^)
α-Fe	2.82 (2.83 *^a^*, 2.87 *^b^*)	22.33 (22.67 *^a^*, 23.64 *^b^*)	205.64 (193.80 *^c^*)	/
NbC	4.49 (4.49 *^d^*, 4.47 *^e^*)	90.77 (90.46 *^d^*, 89.25 *^e^*)	296.40 (302 *^f^*)	−0.70 (−0.73 *^g^*)
TaC	4.58 (4.46 *^d^*, 4.46 *^h^*)	96.03 (88.42 *^d^*, 88.54 *^h^*)	331.30 (318 *^i^*, 342 *^j^*)	−0.75 (−0.76 *^k^*)

*^a^*^, *c*, *g*, *i*, *k*^: theoretical reference value Refs. [15,17,20,21,23]; *^b^*^, *d*, *f*, *j*^: experimental reference value Refs. [16,18,19,22]; *^e^*: experimental reference value; Inorganic Crystal Structure Database (ICSD) # 44496; *^h^*: experimental reference value, ICSD # 43525.

### 3.2. Atom C Mix α-Fe and Model Construction

As the matrix of the composite material in this paper is ferrite, in order to better fit the experimental phenomenon, certain mass fraction of C atoms is doped in Fe bulk phase in the form of interstitial atoms. For BCC configuration α-Fe, there are two kinds of gap positions inside it to accommodate external mix C atoms, which are octahedral gap position and tetrahedral gap position. Beside the octahedral gap center is located at the midpoint of the edge, which is surrounded by six atoms on the top corner of the slightly flattened octahedron. Although the gap radius is shorter than the FCC and BCC gap, only the two nearest atoms need to be moved to expand the lattice in the direction of the nearest atoms, resulting in non-spherically symmetric lattice distortion. Therefore, in this paper C atoms are mixed at the octahedral gap of α-Fe. To ensure that the mass fraction of mix C atoms is within the normal carbon content of ferrite, the (2 × 4 × 2) α-Fe supercell is doped with C atoms at the midpoint of its edge, and final calculated mass fraction of mix C atoms is about 0.9% (Figure 2).

### 3.3. The Surface Characteristics of α-Fe and Carbide MCs

#### 3.3.1. Surface Structure Relaxation

The surface configuration is crucial to the bonding strength of final composite interface, thermodynamic stability, and electronic properties, so the appropriate surface model is import for the high performance of the interface configuration. This paper focuses on low index surfaces, which have more regular atomic structure, thermodynamic stability, and almost no surface reconstruction α-Fe, NbC, and TaC. The low index, closely packed surfaces concerned in this study are (100) surfaces. It is shown from the Figure 3 that the above three surface configurations are stoichiometric surfaces, and the surfaces are terminated by Fe atoms, Nb and C atoms, and Ta and C atoms, respectively.

In the interface configuration, the number of atomic layers needs to be strictly controlled. If the number of atomic layers on the surface is too few, the surface configuration cannot reflect the internal properties, which leads to inaccurate calculation of the mechanical properties. However, if the number of layers is too many, the amount of calculation is extremely large, which leads to a waste of computing resources. Therefore, convergence testing of the surface model is required.

In this paper, Δ*_ij_* represents the change rate of the atomic layer spacing of the surface configuration based on the volume phase spacing. taking the NbC and TaC (100) surfaces as examples, the calculation results of the internal atomic layer displacement are shown in Table 2.

The calculation results show that the atomic relaxation of NbC (100) and TaC (100) surfaces in the geometric optimization process is mainly concentrated in the outermost three atomic layers of each surface configuration, which means that the surface effect is mainly limited to three atomic layers, of which the atomic relaxation of Δ_1,2_ between the outermost and sub-outer layers is the most intense. For the NbC (100) surface, the outermost atoms move strongly toward the interior of the surface during the structural relaxation, and the sub-exoatomic layer (Δ_2,3_ < 0). Similar atomic motion trends are also confirmed on the TaC (100) surface. As shown in Table 2, when the atomic layer thickness on the surface of NbC (100) and TaC (100) reaches 7 layers, the change in the innermost layer spacing can be ignored. At this time the absolute value of atomic displacement on the surface of NbC (100) and TaC (100) decreases to 1.23% and 0.99% respectively. Therefore, symmetrical NbC (100) and TaC (100) surfaces with 7 atomic layers are used in the subsequent interface construction, and the surface configuration is sufficient to show the internal properties of the phase.

#### 3.3.2. Surface Stability

Surface energy describes the energy required to build bulk materials into surface configuration, which can be used to indicate the stability of the surface configuration. The lower the surface energy is, the more stable the surface configuration. At the same time, the convergence of surface configuration can also be measured by surface energy. For stoichiometric α-Fe (100), NbC (100), and TaC (100) surfaces, their surfaces’ energy can be calculated by Formula (3)
(3)$$\delta =\frac{E_{\mathrm{slab}}-nE_{\mathrm{bulk}}}{2A},\;$$
where *E*_slab_ is the total energy of surface structure after full relaxation; *E*_bulk_ is the total energy of the bulk configuration; *n* is the multiple of the stoichiometric formula in the surface configuration; *A* is the surface area, and the coefficient 2 indicates that the surface configuration contains two identical surfaces. In order to avoid the linear increase or decrease in the surface energy caused by the calculation error of the total energy of the volume phase, this paper refers to the methods of Fioretini and Methfessel. By fitting the curve of *E*_slab_ changing with N, *E*_bulk_ is characterized by the slope of the curve to achieve convergence, since the doping of C atoms has little effect on the test of α-Fe surface convergence, the calculation of surface energy is mainly aimed at the α-Fe (100) surface without the doping of C atoms.

Table 3 shows that α-Fe (100), NbC (100), and TaC (100) surfaces vary with the number of atomic layers of the surface configuration of surface energy. As shown in the table, when α-Fe, NbC, and TaC (100) surfaces are over 7 layers, the surface energy begins to converge rapidly. Therefore, in order to obtain reasonable thermodynamic energy data, the surface configuration of all (100) surfaces is selected from atomic slices with seven or more layers, which is sufficient to ensure excellent convergence and calculation accuracy and is consistent with the inference of atomic relaxation results in the above surface configurations. In addition, it can be seen from the table that the surface energy of NbC and TaC (100) surfaces is much less than that of α-Fe (100) surface, and the difference between their surface energy is low, which can be ignored, indicating that both NbC and TaC (100) surfaces have high surface stability.

## 4. Interface Calculation Results and Discussion

### 4.1. Construction of Interface Model

The construction of the surface model is the basis of the interface construction. For the construction of the α-Fe (100) surface model after doping C atom, four (100) surface models were constructed according to the different positions of C atoms in the surface configuration, as shown in Figure 4. In an α-Fe (100) surface with 7 atomic layers, C atoms are located at the 1st, 2nd, 3rd, and 4th layers, and the resulting surface configurations are respectively named layer 1, layer 2, layer 3, and layer 4.

For the structure of the interface model, the NbC and TaC (100) surfaces are rotated at a certain angle and then placed between two α-Fe (100) surface slices in order to match the lattice phase relationship. Through the above matching relationship, this study selected the lattice matching relationship with the lowest lattice mismatch, and the final interface matching relationship is: {100}MCs//{100}Fe, <110>MCs//<110>Fe. Taking the Fe/NbC interface as an example, a sandwich-like interface structure model is constructed as shown in Figure 5.

Since there are four different configurations on the α-Fe (100) surface after doping C atom /doped, eight different types of Fe/NbC and Fe/TaC interfaces are constructed in this study. In order to be consistent with the above, the interfaces are denoted as Fe/NbC-layer 1, Fe/NbC-layer 2, Fe/NbC-layer 3, Fe/NbC-layer 4, Fe/TaC-layer 1, Fe/TaC-layer 2, Fe/TaC-layer 3, and Fe/TaC-layer 4 interfaces. The main purpose of this study is to investigate the effect of different positions of doped C atoms on the bonding strength, stability, and electronic structure of the final interface configuration on the α-Fe (100) surface. In order to facilitate the study, only one type of atomic stacking is considered at the interface, that is, the Fe atom at the interface is located at the top of the Nb and Ta atoms on the NbC or TaC surface.

### 4.2. Interface Bonding Work

Interface bonding work refers to the energy required to separate an interface into two free surfaces, which can be used to measure the bonding strength of the interface. The greater the interface bonding work, the higher the bonding strength of the interface. The binding work of all interfaces can be calculated by Formula (4):(4)$$W_{\mathrm{ad}}=\frac{1}{2A}\left(E_{\mathrm{Fe}}+E_{\mathrm{Mcs}}-E_{\mathrm{Fe}/\mathrm{MCs}}\right),\;$$
where *E*_Fe_, *E*_MCs_, and *E*_Fe/MCs_ are the total energy of bulk α-Fe, MCs and Fe/MCs interface, *A* is the interface area, and coefficient 2 indicates that the interface model contains two identical interfaces.

*W*_ad_ can be calculated in two different ways. One interface model is for unoptimized, the other one is for the optimized interface model. For the first case, the total energy of the unoptimized interface configuration at different interface spacing *d*_0_ is calculated (2.4–4 Å), as well as the total energy of the surface configurations of α-Fe (100) and MCs (100), and then the relationship between *W*_ad_ and interface spacing *d*_0_ is obtained, which is called the universal binding energy ((UBER), as shown in Figure 6). Then, the equilibrium interface spacing *d*_0_ and the equilibrium interface bonding work *W*_ad_ are obtained by fitting the general binding energy curve by the least square method. For the second case, the interface geometry optimized by UBER is used first to relax the interface geometry on the basis of the equilibrium interface spacing *d*_0_, and then the total energy of the relaxed interface configuration is calculated, and finally the *W*_ad_ of the relaxed interface and the corresponding interface spacing *d*_0_ are obtained. By using the first method, the interface bonding work before geometric optimization of all interface configurations can be calculated, which is called the optimal interface spacing and bonding work before geometric optimization. The second method can obtain the spacing and bonding work of the interface configuration in the equilibrium state after geometric optimization, which is called the optimal interface spacing and bonding work after geometric optimization.

The calculation results of interface equilibrium spacing d0 and interface bonding work Wad before and after geometric optimization of all interface configurations are shown in Table 4. It can be seen from the table that before geometric optimization, the bonding strength of Fe/TaC is higher than that of Fe/NbC interface, but after geometric optimization, the Fe/NbC interface has higher bonding work and the interface bonding strength is much better than that of Fe/TaC interface. At the same time, for Fe/NbC and Fe/TaC interfaces, when the doped C atom is located at the layer 2 position in α-Fe, the bonding strength of the interface configuration is significantly better than other interface configurations.

In addition, it can be seen from the table that for Fe/NbC-layer 1 and most Fe/TaC interfaces, the equilibrium spacing of the interface after geometric optimization is larger than that before geometric optimization. This is because the doping of C atoms in α-Fe makes Fe–C atoms bond with each other, electron transfer occurs, and ion bonds with high mutual attraction are formed between Fe–C atoms, which leads to the relaxation of Fe atoms into α-Fe bulk phase, thereby expanding the equilibrium spacing of the composite interface and reducing the bonding strength and stability of the interface. The bonding strength of Fe/NbC-layer 1 and Fe/TaC-layer 1 interface configurations is the worst. When the doped C atom is located at the layer 2 position, the equilibrium spacing of the interface configuration is smaller than that of other interface configurations and has a higher bonding strength. Therefore, in order to achieve high strength bonding at the interface of carbide/iron matrix composites, compared with TaC (100) surface, it is best to use the NbC (100) surface as the first layer atom of the composite interface and locate the doped C atom at the layer 2 position in the α-Fe surface configuration for a higher bonding strength.

### 4.3. Interface Fracture Mechanism of Composites

The fracture properties of the composites mainly depend on the mechanical properties of the constituent phases and the interface structure. When the interfacial strength of the composite interface is greater than the bonding work of its constituent phases, the composite interface belongs to a strong bonding interface, and the cracks and fractures of the material are not easy to appear first at the composite interface and vice versa. 

According to Griffith fracture theory: if G > *W*_ad_, fracture occurs at the interface; if G < *W*_ad_, fracture occurs in the volume phase. The binding work of bulk materials under a certain crystal plane can be calculated by the G~2δ relationship, where δ is the surface energy. For α-Fe, NbC, and TaC (100) surfaces, the surface energies are 2.45 J/m^2^, 1.28 J/m^2^, and 1.27 J/m^2^, respectively, so the bulk binding energies are 4.9 J/m^2^, 2.56 J/m^2^, and 2.54 J/m^2^, respectively.

Comparing the bulk bonding work with the interfacial bonding work listed in Table 4, it can be seen that the interfacial bonding strength of Fe/TaC interface is less than that of bulk Fe and TaC. Therefore, the Fe/TaC interface belongs to the weak bonding interface, and the mechanical failure and fracture of the composite occur preferentially at the interface. For the Fe/NbC interface, when the doped C atoms are located in the second, third, and fourth layers of α-Fe, the interfacial bonding strength of the composite is higher than that of the bulk NbC, and the interface belongs to the strong bonding interface. According to Griffith fracture theory, cracks in the composite first initiate and expand at the bulk NbC rather than the interface.

### 4.4. Interface Electronic Structure

Generally, the mechanical properties of the interface are closely related to the bonding characteristics of the interface atoms. Figure 7 shows the differential electron density distribution of the Fe/NbC-layer 2 and Fe/TaC-layer 2 interface configurations (unit: electron/Å^3^), which can be used to determine the bonding characteristics of the interface. The red part of the figure represents the loss of electrons, while the blue part represents the gain of electrons. For Fe/NbC and Fe/TaC interfaces, it can be seen from the diagram that electrons are concentrated between Fe–Nb and Fe–Ta at the interface, and there is basically no charge transfer between atoms, and Fe–Nb and Fe–Ta metal bonds are formed at the interface. At the same time, there is high electron density and obvious charge transfer between Fe–C, Nb–C, and Ta–C atoms in bulk Fe, NbC, and TaC, in which Fe, Nb, and Ta atoms obtain electrons, C atoms lose electrons, ion bonds are formed between the bulk phases in the composites, and the bonding strength is high.

In addition, due to the distance between Fe atoms and Nb atoms at the Fe/NbC interface is the closest to (Å) compared with Fe atoms and Ta atoms, the degree of charge transfer between Fe–Nb atoms is greater than that of Fe–Ta atoms, and the metal bond force between the two atoms is stronger, the bonding strength of the Fe/NbC-layer 2 interface is higher than that of the Fe/TaC-layer 2 interface. This is consistent with the conclusion of the above interface bonding work.

Finally, this paper models and calculates based on the first principle, and then optimizes the modeling model according to the experimental results, so as to more accurately control the interface structure of composite materials.

## 5. Conclusions

The interfacial bonding strength, fracture mechanism, and electronic structure of α-Fe composites reinforced by transition metal carbide particles (NbC and TaC) are systematically studied by first-principles based on density functional theory. Considering the different positions of doped C atoms in α-Fe, eight kinds of Fe/MCs interfaces are calculated in this paper. The calculation results of surface energy show that the difference of surface energy between NbC (100) and TaC (100) is little, and both have high surface stability.
(1)For the Fe/MCs (M = Nb, Ta) interface, the bonding strength of the Fe/NbC interface is higher than that of the Fe/TaC interface, and the bonding strength is the highest when the doped C atom is located at the layer 2 position in the α-Fe surface configuration;(2)By analyzing the fracture mechanism of the composite interface, it is found that the Fe/TaC interface is a weak bonding interface, and the mechanical failure and fracture of the composite material preferentially occur at the interface. The bonding strength of the Fe/NbC interface is generally higher than that of the bulk NbC, and the crack first initiates and propagates at the bulk NbC rather than the interface.(3)The difference electron density map shows that the metal/ion mixed bonds are formed in the Fe/MCs interface region, and the atomic spacing in the Fe/NbC interface is short, and the interface bonding strength is high.

## Figures and Tables

**Figure 1 materials-16-01887-f001:**
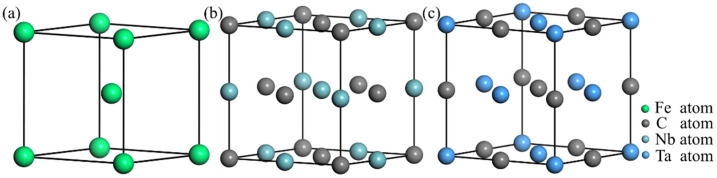
Crystal structure of α-Fe and MCs: (**a**) α-Fe; (**b**) NbC; (**c**) TaC.

**Figure 2 materials-16-01887-f002:**
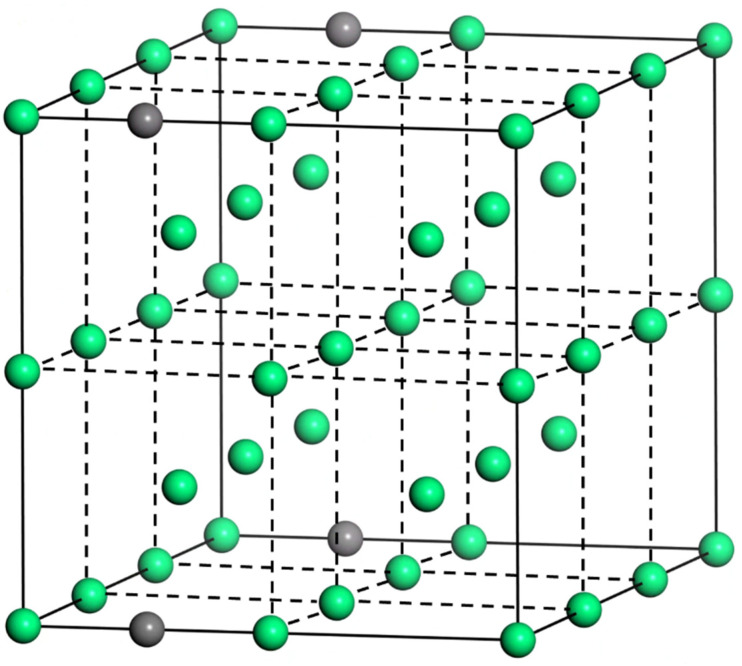
Schematic diagram of crystal structure of Fe mix with C atom.

**Figure 3 materials-16-01887-f003:**
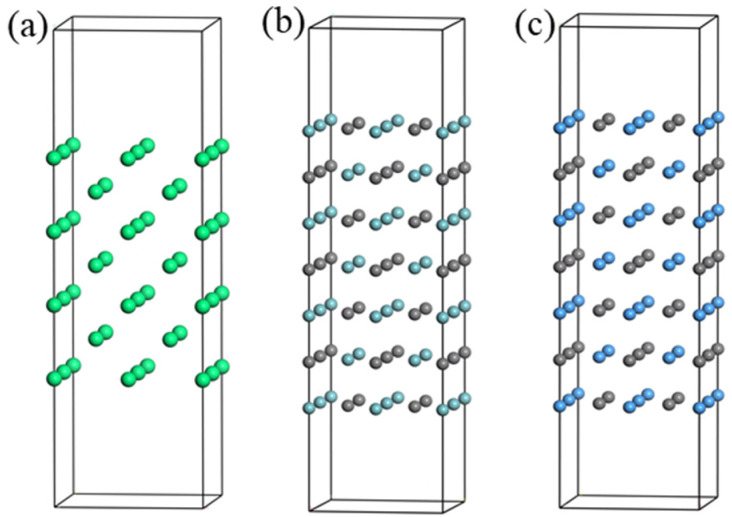
α-Fe, NbC, and TaC low-index surfaces of crystal structure. (**a**) α-Fe(100): (**b**) NbC (100): (**c**) TaC(100) surface.

**Figure 4 materials-16-01887-f004:**
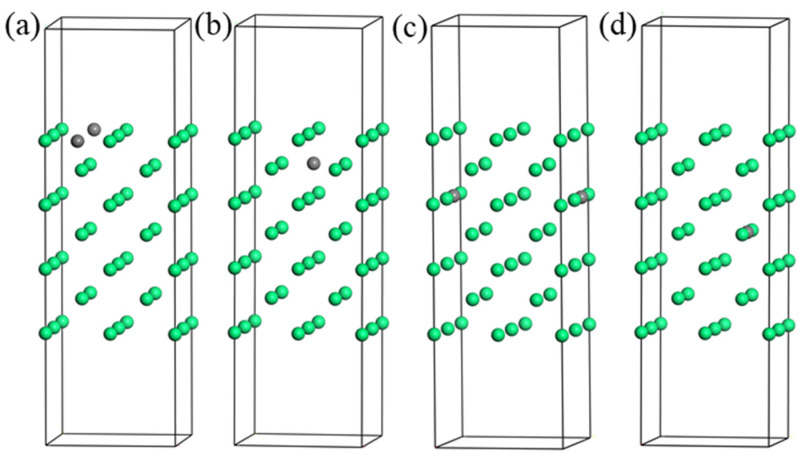
The atomic structure diagram of α-Fe (100) surface configuration doped with C atoms at different positions: (**a**) layer 1; (**b**) layer 2; (**c**) layer 3; (**d**) layer 4.

**Figure 5 materials-16-01887-f005:**
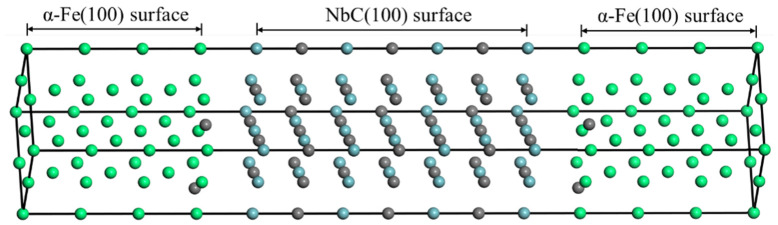
Interface structure of Fe/NbC composite.

**Figure 6 materials-16-01887-f006:**
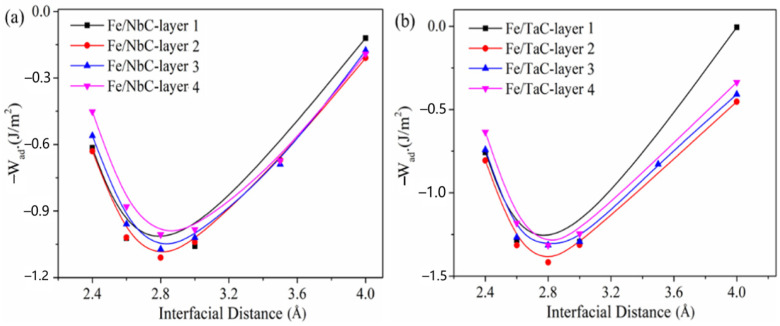
General binding energy curves of Fe/NbC and Fe/TaC interface configurations before geometric optimization (the minimum value of the curve represents the equilibrium spacing of α-Fe and MCs slices): (**a**) Fe/NbC; (**b**) Fe/TaC interface.

**Figure 7 materials-16-01887-f007:**
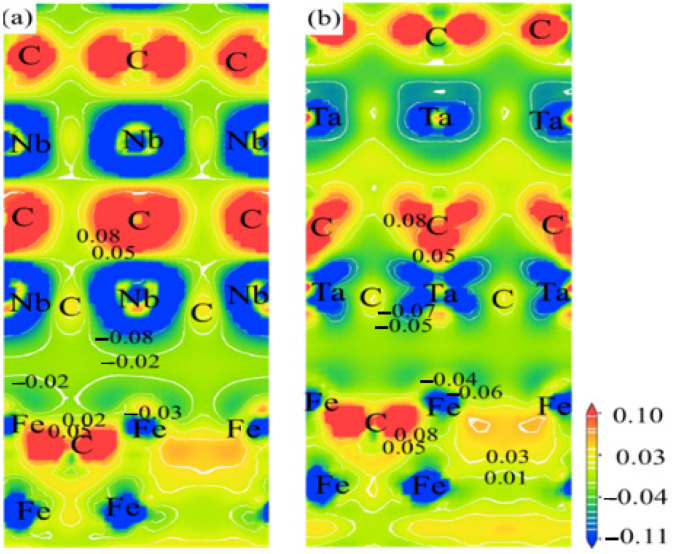
A long (100) crystal phase (**a**) Fe/NbC-layer 2; (**b**) differential electron density distribution at the Fe/TaC-layer 2 interface.

**Table 2 materials-16-01887-t002:** Variation rate of atomic layer displacement relative to volume phase spacing on NbC and TaC (100) surfaces with slice thickness.

Surface	Termination	Interlayer	Slab Thickness (*n*)			
			5	7	9	11
NbC (100)	Nb&C	Δ_1,2_	−3.30	−2.87	−1.98	−1.65
		Δ_2,3_	−2.44	−2.32	−2.28	−2.12
		Δ_3,4_		−1.23	−0.95	−0.41
		Δ_4,5_			1.22	0.87
		Δ_5,6_				−0.36
TaC (100)	Ta&C	Δ_1,2_	−3.17	−3.12	−3.13	−3.12
		Δ_2,3_	−1.41	−1.25	−1.11	−1.11
		Δ_3,4_		−0.99	−0.90	−0.76
		Δ_4,5_			−0.80	−0.71
		Δ_5,6_				−0.67

**Table 3 materials-16-01887-t003:** Surface energy of α-Fe, NbC (100) and TaC (100) surfaces with different atomic layers.

Number of Layers (*n*)	Surface Energy (J/m^2^)		
	α-Fe (100)	NbC (100)	TaC (100)
3	2.47	1.29	1.27
5	2.46	1.29	1.27
7	2.45	1.28	1.26
9	2.45	1.28	1.27
11	2.45	1.28	1.27

**Table 4 materials-16-01887-t004:** Equilibrium spacing (*d*_0_) and bonding work (*W*_ad_) of Fe/MCs interface before and after geometric optimization).

Interfaces	Layer	Unrelaxed		Relaxed	
		*d*_0_ (Å)	*W*_ad_ (J/m^2^)	*d*_0_ (Å)	*W*_ad_ (J/m^2^)
Fe/NbC	1	2.78	1.01	3.32	2.00
	2	2.81	1.09	2.61	2.72
	3	2.84	1.05	2.71	2.61
	4	2.86	0.99	2.73	2.60
Fe/TaC	1	2.79	1.25	3.04	1.62
	2	2.8	1.38	2.87	2.10
	3	2.82	1.3	2.97	1.98
	4	2.79	1.28	3.02	1.99

## Data Availability

Only the data of this paper is available.

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
