# Peer review of "Interface Calculation of In Situ Micro-Nano TaC/NbC Ceramic Particle Composites"

_materials, 2023, doi:10.3390/ma16051887_

Round 1

Reviewer 1 Report

Review of the paper title “Interface calculation of in-situ micro-nano TaC/NbC ceramic
particle composites”

Overview

The paper reported the first-principles calculation method used to
systematically study the interface bonding work, interface bonding characteristics, and electronic properties of α-Fe and NaCl-type transition metal carbides (NbC and TaC). The bonding strength of the composite interface system is accurately measured, and the interface strengthening mechanism is analyzed from the perspectives of
atomic bonding and electronic structures. The author claims that this could provide a scientific guiding ideology for regulating the interface structure of composite materials.

The reported work is interesting and original, but numerous problems still can be found. Therefore, I think this paper could be considered for publication after moderate revision. 

1.      The English abbreviations of some nouns were not well-listed when they first appeared in the manuscript, such as NbC and TaC, please well-explain to make it reader-friendly.

2.       Abstract is not well structured; it should be revised while highlighting the method used for the study. The abstract should be systematic, focusing on

a.      General background

b.      Specific background

c.       Knowledge gap

d.      Methos and results etc.

e.      Meaning of results

3.      Section 2, needs some more clarifications, this should be revised by adding more information. For example “The surface and interface models are constructed with periodic boundary conditions’ there should be more information related to the models used.

4.      In my opinion the Limitations of the study should be discussed.

5.      Dislocation core structure between composite phases and interface energy is not discussed it should be discussed in detail.

Final Decision: Major revision is suggested

Author Response

Dear Editors,

We sincerely thank you for your invaluable comments and suggestions on our manuscript entitled “Interface calculation of in-situ micro-nano TaC/NbC ceramic particle composites”( materials-2186053). We appreciate the time and efforts that you dedicated to providing feedback on our manuscript and are grateful for the insightful comments on and valuable improvements to our paper. We have carefully made corrections which we hope meet with

approval.

detailed information see attachment

Sincerely yours,

Jilin Li

Feb. 6, 2023

Reviewer 2 Report

The article "Interface calculation of in-situ micro-nano TaC/NbC ceramic particle composites" is related to the modeling and theoretical study of interfaces. The research is stated to be theoretical. For a correct perception of the article it is necessary to confirm experimentally the modeling of the structure and properties of the interfaces. The Abstract states that it is difficult to accurately measure the interfacial properties of composites by traditional experiments. However, there are modern methods and approaches that allow to solve these problems. First of all, this is the method of transmission electron microscopy, which allows one to study the structure at the atomic level. There are striking examples of studies of multilayer X-ray mirrors (e.g., W/Si) deposited by magnetron sputtering. In this case, the method of transmission electron microscopy allows to visualize the real atomic structure in W and Si layers, as well as to observe the interfacial boundary WSi. And there are a lot of such examples. A number of other interesting methods are also suitable for the study of layered structures and interphase boundaries. First of all, it is a method of X-ray reflectometry, which allows to study under the total external reflection conditions the reflection curve and the thickness oscillation of layered structures, determination of interphase boundary and its thickness, densities and roughness of layers. The method of X-ray diffraction in the Bragg diffraction conditions makes can be use to study both the phase composition of crystalline films and to determine their thicknesses and the presence of interphase boundaries. The applicability of X-ray spectroscopy methods (XANES, XPS) should also be mentioned separately, which makes it possible to determine the formation and presence of new chemical bonds at interphase boundaries. Therefore, the content of the article looks incomplete and requires serious revision and revision. There are also more technical inaccuracies in the article: 1. The numbering of the references should not be superscript [1], but [1]. 2. Formulas (1) and (2) must not have a ledge, each must be presented on one line, separated from each other by a comma. Formula (2) should also be followed by a comma, and the next line should start without indentation. Similarly, formulas (3) and (4) should have a comma after the expressions, and the next line should start without indentation, and in expression (4) the line should start with a capital letter. 3. Part 3.2. Atom C mix α-Fe and model construction 4. Page 3, line 1 from top: (C11=578.53 GPa, C12=160.17 GPa, C44=155.77 GPa 5. Page 3, line 1 from top: resulting in non spherically symmetric lattice distortion. 6. Page 3, line 23 from top: about 0.9% (Fig. 2). 7. Page 3, line 11 from bottom: in Table 2. 8. Page 4, line 5 from top: 3.2.2. Surface stability 9. Page 4, line 39 from top: uration, as shown in Fig. 4. 10. Page 4, line 41 from top: tions are respectively named layer 1, layer 2, layer 3 and layer 4. 11. Page 4, line 46 from top: terface matching relationship is: {100}MCs//{100}Fe, MCs//Fe. 12. Page 4, line 48 from top: as shown in Fig. 5. 13. Page 5, lines 1-3 from top: study. In order to be consistent with the above, the interfaces are denoted as Fe/NbC-layer 1, Fe/NbC-layer 2, Fe/NbC-layer 3, Fe/NbC-layer 4, Fe/TaC-layer 1, Fe/TaC-layer 2, Fe/TaC-layer 3 and Fe/TaC-layer 4 interfaces, respectively. 14. Page 5, line 22 from top: universal binding energy (UBER, as shown in Fig. 6). 15. Page 5, line 8 from bottom: In addition, it can be seen from the table that for Fe/NbC-layer 1 and most Fe/TaC 16. Page 5, line 1 from bottom: strength of Fe/NbC-layer 1 and Fe/TaC-layer 1 interface configurations is the worst. When 17. Page 6, line 1 from top: the doped C atom is located at the layer 2 position, the equilibrium spacing of the interface 18. Page 6, line 6 from top: atom at the layer 2 position in the α-Fe surface configuration for a higher bonding strength. 19. Page 6, lines 16-18 from top: energy. For α-Fe, NbC and TaC(100) surfaces, the surface energies are 2.45 J/m2 , 1.28 J/m2 and 1.27 J/m2 , respectively, so the bulk binding energies respectively are 4.9 J/m2 , 2.56 J/m2 and 2.54 J/m2 . 20. Page 6, lines 30-33 from top: ing characteristics of the interface atoms. Fig. 7 shows the differential electron density distribution of the Fe/NbC-layer 2 and Fe/TaC-layer 2 interface configurations (unit: electron/Å3 ), which can be used to determine the bonding characteristics of the interface. The 21. Page 6, line 47 from top: higher than that of the Fe/TaC-layer 2 interface. This is consistent with the conclusion of 22. Page 7, line 5 from top: (1) For the Fe/MCs (M=Nb, Ta) interface, the bonding strength of the Fe/NbC inter23. Page 7, line 7 from top: when the doped C atom is located at the layer 2 position in the α-Fe surface configuration. 24. Page 8: Table 2. Variation rate of atomic layer displacement relative to volume phase spacing on NbC and TaC (100) surfaces with slice thickness Slab thickness (nm) 25. Page 8: Table 3. Surface Energy of α-Fe, NbC and TaC (100) Surfaces with Different Atomic Layers 26. Page 9: Table 4. Equilibrium spacing (d0) and bonding work (Wad) of Fe/MCs interface before and after geometric optimization) Fig. 1. Crystal structure of α-Fe and MCs : (a) α-Fe ; (b) NbC ; (c) TaC. Fig. 2. Schematic diagram of crystal structure of Fe mix with C atom. Fig. 3. α-Fe, NbC and TaC low-index surfaces of crystal structure. Fig. 4. The atomic structure diagram of α-Fe (100) surface configuration doped with C atoms at different positions: (a) layer 1; (b) layer 2; (c) layer 3; (d) layer 4. Fig. 5. Interface structure of Fe/NbC composite. Fig. 6. General binding energy curves of Fe/NbC and Fe/TaC interface configurations before geometric optimization(the minimum value of the curve represents the equilibrium spacing ofα-Fe and MCs slices): (a) Fe/NbC; (b) Fe/TaC interface. In the figure 6 you should also change the caption. It is necessary to replace “layer1” by “layer 1” ….. Fig. 7. A long (100) crystal phase (a) Fe/NbC-layer 2; (b) differential electron density distribution at the Fe/TaC-layer 2 interface.

Author Response

(The authors gave the same response as above.)

Reviewer 3 Report

See comments in Referee Report file

Author Response

(The authors gave the same response as above.)

Round 2

Reviewer 1 Report

The author has incorporated the suggested corrections so i feel pleasure to accept the manuscript for publication

Author Response

Dear Editors,

We sincerely thank you for your approval.

If you have any question, please contact me

Sincerely yours,

Prof. Jilin Li

Feb. 10, 2023

Reviewer 2 Report

Minor revision:

ABSTRACT

(1) “[NbC(Niobium Carbide) and TaC(Tantalum Carbide) ].” replaced by (NbC (Niobium Carbide) and TaC (Tantalum Carbide)).

1. Introduction

Page 1

(1) “conductivity[1-3]” replaced by “conductivity [1-3]”

(2) “Park et al[4]” replaced by “Park et al [4]”

(3) “Jung et al[5]” replaced by “Jung et al [5]”

(4) “Chung et al[1]” replaced by @Chung et al [1]”

(5) ”phase[5].” replaced by ”phase [5].

Page 2

(1) “carbides[6-8],” replaced by “carbides [6-8],

(2) “ship[9,10]:{100}MCS//{100}Fe,” replaced by “ship [9,10]: {100}MCS//{100}Fe,”

2. Methods and details

(1) “in this study[11]” replaced by “in this study [11]”

(2) “cores[12]” replaced by “cores [12]

(3) “state[13]” replaced by “state [13]”

(4) “conditions[24]” replaced by “conditions [24]

(5) method[14]” replaced by “method [14]

3.1. Bulk phase characteristics of α-Fe and carbide MCs

(1)

replaced by

, (1)

, (2)”

(2) values[15-23]” replaced by “values [15-23]

Page 4

(1) “3.3.2. surface stability” replaced by “3.3.2. Surface stability”

(2) (3),” replaced by , (3)”

(3) “Where Eslab” replaced by “where Eslab

Page 5

(1) “ (4),” replaced by “, (4)”

Page 7

(1) “Fe/NbC-layer2” replaced by “Fe/NbC-layer 2”

Author Response

Dear Editors,

We sincerely thank you for your invaluable comments and suggestions on our manuscript entitled “Interface calculation of in-situ micro-nano TaC/NbC ceramic particle composites”( materials-2186053). We appreciate the time and efforts that you dedicated to providing feedback on our manuscript and are grateful for the insightful comments on and valuable improvements to our paper. We have carefully made corrections which we hope meet with

approval.

detailed information see attachment

Sincerely yours,

Prof. Jilin Li

Feb. 10, 2023
